# Rough set based information theoretic approach for clustering uncertain categorical data

**Jamal Uddin** [1]☯*, **Rozaida Ghazali** [2]☯, **Jemal H. Abawajy** [3]‡, **Habib Shah** [4]‡, **Noor Aida Husaini** [5]☯, **Asim Zeb** [6]‡

1 Qurtuba University of Science & IT, Peshawar, Pakistan, 2 Universiti Tun Hussien Onn Malaysia, Batu Pahat, Johor, Malaysia, 3 Deakin University, Waurn Ponds Campus, Geelong, Australia, 4 King Khalid University, Asir, KSA, 5 Tunku Abdul Rahman University College, Kuala Lumpur, Malaysia, 6 Abbottabad University of Science & Technology, Abbottabad, Pakistan

☯ These authors contributed equally to this work.
‡ JHA, HS and AZ also contributed equally to this work.
* jamuddin1983@qurtuba.edu.pk

**Data Availability Statement:** The data from this study is third-party. Those interested can access these data in the same manner as the authors. The authors did not have any special access privileges

## Abstract

### Motivation

Many real applications such as businesses and health generate large categorical datasets with uncertainty. A fundamental task is to efficiently discover hidden and non-trivial patterns from such large uncertain categorical datasets. Since the exact value of an attribute is often unknown in uncertain categorical datasets, conventional clustering analysis algorithms do not provide a suitable means for dealing with categorical data, uncertainty, and stability.

### Problem statement

The ability of decision making in the presence of vagueness and uncertainty in data can be handled using Rough Set Theory. Though, recent categorical clustering techniques based on Rough Set Theory help but they suffer from low accuracy, high computational complexity, and generalizability especially on data sets where they sometimes fail or hardly select their best clustering attribute.

### Objectives

The main objective of this research is to propose a new information theoretic based Rough Purity Approach (RPA). Another objective of this work is to handle the problems of traditional Rough Set Theory based categorical clustering techniques. Hence, the ultimate goal is to cluster uncertain categorical datasets efficiently in terms of the performance, generalizability and computational complexity.

### Methods

The RPA takes into consideration information-theoretic attribute purity of the categorical-valued information systems. Several extensive experiments are conducted to evaluate the

that others would not have. The data can be accessed via the following links: 1. Balloons (https://archive.ics.uci.edu/ml/datasets/balloons) 2. Car Evaluation (https://archive.ics.uci.edu/ml/datasets/car+evaluation), 3. Zoo (https://archive.ics.uci.edu/ml/datasets/zoo) 4. Chess (https://archive.ics.uci.edu/ml/datasets/Chess+(King-Rook+vs.+King-Pawn)), 5. Balance scale (https://archive.ics.uci.edu/ml/datasets/balance+scale), 6. Monk's problems (https://archive.ics.uci.edu/ml/datasets/MONK's+Problems).

**Funding:** The authors would like to thank the King Khalid University of Saudi Arabia for supporting this research under grant number R.G.P.1/365/42.

**Competing interests:** The authors have declared that no competing interests exist.

efficiency of RPA using a real Supplier Base Management (SBM) and six benchmark UCI datasets. The proposed RPA is also compared with several recent categorical data clustering techniques.

## Results

The experimental results show that RPA outperforms the baseline algorithms. The significant percentage improvement with respect to time (66.70%), iterations (83.13%), purity (10.53%), entropy (14%), and accuracy (12.15%) as well as Rough Accuracy of clusters show that RPA is suitable for practical usage.

## Conclusion

We conclude that as compared to other techniques, the attribute purity of categorical-valued information systems can better cluster the data. Hence, RPA technique can be recommended for large scale clustering in multiple domains and its performance can be enhanced for further research.

## 1 Introduction

Advances in computational, faster and cheaper storage and communication technologies have led to the generation and storage of very large and complex data by businesses, governmental agencies and other organizations. The collected data can be used for important business decisions such as better understanding market dynamics, customers spending trends, operations and internal business processes. However, size and complexity of the data render it beyond the ability of a human analyst to process it for the purpose of decision making process. Similarly, in these processes, the issue of uncertain attribute value appears as a result of instrument fault, approximations in measurement or even subjective by assessments expert etc [1]. Moreover, as much of the data is uncertain and categorical in nature, it poses defiance to the conventional data analytic approaches. As a result, there is a surge of interest in methods for mining uncertain categorical data recently [2–5]. Discovering useful knowledge these data sets efficiently is a serious requirement and a huge economic need.

Clustering a set of objects into homogeneous groups is a fundamental operation in data mining. Clustering methods are often used to support data-driven decision making in numerous domains such as Businesses (e.g., market dynamic analysis) [6], Healthcare (e.g., protein sequence analysis) [7–9], Science (e.g., environmental data analysis) [10], Information Security [11], Computer Networks [12], Image Segmentation [13] and Software Maintenance [14, 15]. In data analytics, clustering method lies at the core of successful data analysis tasks such as data summation, classification as well as data reduction, filtering, exploratory data analysis and many more [14, 16–19]. A variety of cluster analysis methods for numerical data analysis are commonly deployed by organizations. These cluster analysis methods are not appropriate for categorical dataset processing. The increasing proliferation of large uncertain categorical data sets poses significant challenges to the contemporary clustering techniques.

Recently, attention has been put on data with non-numerical attributes or categorical attributes. There have been progresses in categorical data clustering [20–24]. Although these clustering methods show advancement in categorical data clustering and analysis, they are not suitable for uncertain categorical datasets and suffer from stability issues [25]. Recently,

approaches that are based on fuzzy sets [20, 26–28] and Rough Set Theory (RST) [25, 29–33] for clustering categorical data have appeared in the literature. However, fuzzy sets based methods require heavy computational complexity as they require several runs each time with new initial value to assess the clustering outcome stability. Moreover, a parameter that controls the membership fuzziness need to be adjusted to achieve better clustering results.

In the process of dealing with categorical data and handling uncertainty, the Rough Set Theory has become well-established mechanism in a wide variety of applications including databases. Two types of uncertainty can be modeled by Rough Set Theory inherently [34–36]. The indiscernibility relation gives rise to the first type of uncertainty. The indiscernibility relation partitions all values into a finite set of equivalence classes and is imposed on the universe. The second type of uncertainty is modeled through the approximation regions in Rough Sets. Here, the elements of upper approximation region have uncertain participation, whereas the lower approximation region have total participation.

Rough Set Theory (RST) is a mathematical concept to imperfect analysis. It was discussed in greater detail in [12, 30]. The RST is a viable system to deal with uncertainty in clustering process of categorical data. RST was originally a symbolic data analysis tool now being developed for cluster analysis. RST clusters the universe, and describe its subsets as classes of equivalence. It also helps in decision making on uncertain data [31]. For example, symptoms form information about patients of a certain disease. In view of their available symptoms, the similar or indiscernible patients are characterized by the same symptoms. This way of generating the indiscernibility relation is the mathematical basis of Rough Set Theory.

Maximum Dependence Attribute (MDA), Maximum Significance of Attribute (MSA), Information Theoretic Dependency Roughness (ITDR) and other recent rough set based techniques [31–33] outperformed their predecessors [25, 37] for clustering categorical data. However, these recent techniques suffer from low accuracy, high computational complexity and generalizability issues especially on data sets where they sometimes fail or hardly select their best clustering attribute. Some of their limitations are outlined:

1. MDA technique cannot perform well on data sets with attributes having zero or equal dependency value.

2. MSA technique also fails to select clustering attribute on data sets having attributes with zero or equal significance value.

3. ITDR techniques face issues like random attribute selection and integrity of classes due to presence of entropy measure.

Hence, an efficient technique is needed to cluster uncertain categorical datasets in terms of the accuracy, generalizability and computational complexity. In this paper, we propose a new information theoretic Rough Purity Approach (RPA) for categorical data clustering that addresses the problems inherent in the existing RST based clustering techniques. RPA utilizes the Rough Attribute Dependencies based on purity measure [38–41] in categorical valued information systems. The representation of uncertain information by purity has been applied to all areas of databases, including data mining [39], knowledge extraction [40], cluster validation [42] and information retrieval [41]. Hence, this paper relates the concept of information theoretic purity to Rough Sets to establish a new Rough Set metric of uncertainty which is Rough Purity. A Supplier Base Management real data set and several UCI benchmark data sets are used to validate the effectiveness of the proposed approach [43].

The Accuracy, Entropy, Purity, Rough Accuracy, Iterations and Time are some measures to test the quality of the obtained clusters. Moreover, validating the clustering results is a non-trivial task. The ratio of correctly clustered and total objects gives Accuracy [44]. The degree to

which each cluster consists of objects from a single class is called entropy and better clustering performance has smaller entropy [39, 45]. The extent to which a cluster contains objects of a single class is known as Purity measure [39]. A better clustering result must have high overall purity and a value of 1 shows perfect clustering. The mean roughness of selected clustering attribute will give the Rough Accuracy. Higher mean Roughness implies better accuracy [31].

The computational complexity of clustering task can be determined by number of iterations required for finding the indiscernibility relations. It also includes finding the maximum or minimum values of dependence, significance, Rough Entropy, Rough Purity etc. The computational complexity of any technique can also be illustrated in terms of respond time. Here, the response time of CPU in milliseconds is counted to examine the performance of clustering task. A better technique in terms of response time will always consume less time.

The rest of this paper is organized as follows. Section 2 describes the overview of the related work in the field of Cluster Analysis, Rough set theory, and categorical data clustering. To explore the limitations of Rough categorical clustering techniques, the analysis of existing techniques on an illustrative example is presented in Section 3. Section 4 introduces the concept of a new and proposed information theoretic Rough Purity measure. An illustrative example and proposition illustrating the methodology and significance of proposed approach is also highlighted. The experimental setup and data sets is described in Section 5. The experiments and the discussion on results are presented in Section 6. The summary of results and threats to validay are discussed in Section 7 and Section 8 respectively. Section 9 concludes the article at the end.

## 2 Related work

### 2.1 Cluster analysis

Clustering is a summary and generative or concise model of the data without explicit labels. The basic issue of clustering is splitting the data objects into potential similar sets. There are significant variations in this issue depending on clustering model and data type. The clustering methods are utilized to support data-driven decision making in many domains such as software maintenance, information security, science, businesses and health care [29]. The application areas in which the clustering is required are social network analysis, biological data analysis, multimedia data analysis, dynamic trend detection, data summarization, customer segmentation and collaborative filtering [46]. Moreover, it is also utilized as intermediate step for other fundamental data mining problems. A wide variety of cluster analysis techniques is employed to address the clustering problems [42, 47]. The commonly used clustering techniques include Feature Selection Methods, Probabilistic and Generative Models, Distance-Based Algorithms, Density and Grid-Based Methods, Leveraging Dimensionality Reduction Methods, Model-based Methods, Matrix Factorization and Co-Clustering, Spectral Methods [17].

The existing work on cluster analysis techniques is summarized in Table 1.

### 2.2 Rough Set Theory

The uncertain categorical data is used in several areas nowadays and the classical clustering methods are unable to handle such data. Accordingly, several uncertain categorical clustering methods got attention. Pawlak in 1982 introduces Rough Set Theory (RTS) which is an approach to deal with uncertainty and vagueness. The RST has appeared as an essential concept for dealing with different tasks like identifying and evaluating data dependency, reasoning of uncertain data and reduct of information. Moreover, it is useful for representing and

**Table 1. Summary of related work on cluster analysis.**

| Paper | Proposed Technique | Compared Techniques | Evaluation Metrics | Data Sets/ Application Area |
|---|---|---|---|---|
| [48] | Fuzzy Cluster Analysis | Fuzzy C-Means | Consensus threshold, Time of Iterations, Number of Clusters | Emergency Response Plan Selection |
| [49] | New strategy for cluster analysis | Network-determined mechanisms | Polarity, Correlation | Focal mechanism |
| [45] | Clustering Based on Entropy (CBE) | K-means, fuzzy c-means, Bayes classifier, Multilayer perceptron | Effectiveness | Synthetic Gaussian and non-Gaussian datasets, UCI datasets |
| [18] | Agglomeration methods | K-means | Inclusiveness, contestation | Political Science |
| [17] | Taxonomy and empirical analysis | Classical clustering algorithms | Stability, runtime, and scalability tests | MHORD), MHIRD, SHORD, SHIRD, SPFDS, DOSDS, SPDOS, WTP, DARPA, ITD B Big data sets |
| [50] | Survey | Partition based Clustering Algorithms | Number of clusters | Medical data sets |
| [51] | Cooperative clustering technique | Agglomerative, LIMBO, Wcombined | MoJoFM measure, arbitrary decisions | Object oriented software systems, Mozilla |
| [52] | Empirical study | Several clustering methods | Segmentation Variables, Number of clusters | Marketing research |
| [53] | Combined and Weighted Algorithms | Agglomerative approaches | Arbitrary decisions, Number of clusters | Open source software systems written |
| [54] | Refined rough cluster algorithm | Rough cluster algorithm | Objective function, stability | Synthetic, forest and gene data. |
| [47] | Survey | Several clustering algorithms | Percentage error, Accuracy | Iris, Mushroom, Salesman problem, Bio-informatics. |
| [55] | Self-Splitting and Competitive Learning | OPTOC | Number of clusters | Gene Expression Data |
| [56] | Segmentation and phantom study | Manual ROI | Average mean squared error, time | PET Images, lung data |
| [23] | CACTUS | STIRR | Similarity, time | Real and synthetic datasets |
| [57] | Software Re-modularization | Complete, single, weighted | Precision, Recall, Cohesion, Coupling, Similarity | gcc, Linux, Mosaic and real world legacy system |
| [20] | Extented k-Means and k-modes | k-Means and k-modes | Accuracy, run time, standard deviation | Soybean disease and credit approval |
| [58] | Decision support approach | Average linkage, Centroid, Ward's | Growth rate, Gamma frequency | Large scale R and D planning. |
| [59] | Fine-classification procedure | Cluster classification | Spectra | Land and marine object |
| [60] | Silhouettes | Fuzzy clustering | Average silhouette width, Number of clusters | Ruspini |

analyzing the uncertain, vague and imprecise knowledge, data patterns and accessibility of consistent information [30].

In RST, the viewpoint is that every object of the universe has associated some information (knowledge, data) and the objects are similar or indiscernible characterized by the identical information. Accordingly, an indiscernibility relation is generated in this way which is the fundamental mathematical concept of RST. This relation somehow resembles with Leibniz's Law of Indiscernibility. The rough indiscernibility relations are developed in context of an arbitrary set of attributes. Other data analysis tool need additional information like basic probability assignments in Dempster–Shafer theory, probability distributions in statistics and grade of membership of fuzzy set theory whereas the RST does not have any such requirement about data hence it is better. The precise concepts in contrast to vague concepts can be characterized in terms of information about the objects. Accordingly, as pair of precise concepts the RST replaces any vague concept by an upper and lower and approximation. All possibly belonged objects for each concept are included in upper approximation whereas all surely belonged

objects are in lower approximation. A boundary region of any concept is the difference of upper and lower and approximation. Hence, despite of membership of a set a boundary region is employed in RST to express the vagueness [12].

The boundary region of a set is non-empty when the knowledge about set is not enough to describe the set precisely. Therefore, a set having empty boundary region is crisp otherwise it is rough. This idea of vagueness resembles exactly that is proposed by Frege [61] whereas the lower and upper and approximations of a set coincides with the interior and closure operations of topology [62]. Different effective RST based techniques were developed for exploring hidden patterns and determining optimal sets in data. Moreover, it assists in evaluating the data significance and developing the decision rules from data [31]. The RST utilized in numerous applications by researchers which is summarized in Table 2.

## 2.3 Categorical data clustering

The classical techniques for clustering are limited for numeric data however, the categorical data is multi-valued and similarity may be termed as identical objects, values or both. In categorical type of data, the tables with fields are not naturally illustrated by a metric for example certain symptoms of a patient, names of automobiles producers and manufacturer products. Therefore, the clustering of categorical data is more challenging as there is no inherent distance measure. Though, several valuable categorical clustering algorithms are introduced but they are not designed to deal with uncertainty [31]. Accordingly, the clustering of categorical data where no sharp boundary is present between clusters rises as an important problem of the real world applications.

This uncertainty in categorical data clustering is handled using fuzzy sets where the clusters of categorical data is represented with fuzzy centroids [26]. The fuzzy set based algorithm and conventional algorithms are tested and compared on some categorical clustering data sets. Though, better performance is obtained by the fuzzy set based algorithm but to get a satisfactory value for even one parameter it requires multiple runs. Similarly, to achieve stability the fuzzy membership need to be controlled.

Some substantial contributions are offered by rough set based techniques which handles uncertainty and cluster categorical data. The rough set based Total roughness (TR) and Bi-clustering (BC) techniques select best clustering attribute and handle the uncertainty issue [37]. The BC technique is limited to bi-valued attributes whereas the TR works on multiple-valued attributes. Moreover, the limited data, arbitrarily selection and imbalance clustering are key limitations of both techniques.

Min–Min-Roughness (MMR) is another rough set based clustering technique for categorical data having the significant ability to handle uncertainty by user itself [25]. The MMR technique outperforms against K-modes, fuzzy K-modes and fuzzy centroids on Zoo and Soybean data. The proposed technique is also tested against ROCK, Squeezer, hierarchical and other algorithms on comparatively larger date of Mushroom data. The stable results of MMR technique are subject to number of clusters as input. The MMR clustering technique is modified as MMeR for dealing with uncertainty, numerical and categorical features at the same time [79]. The MMeR has ability to deal with heterogeneous data by generalizing the hamming distance. A new modified hamming distance is accordingly developed for any two data objects. The experimental results show better performance of MMeR as compare to some existing algorithms on several data sets.

Certain limitations related to computational complexity and accuracy of previous techniques were resolved by suggesting an improved rough set based categorical clustering technique named Maximum Dependency Attributes (MDA) [31]. The clustering attribute in

**Table 2. Summary of related work on rough set theory.**

| Paper | Proposed Technique | Compared Techniques | Evaluation Metrics | Data Sets/ Application Area |
|---|---|---|---|---|
| [35] | Integrated Fuzzy PIPRECIA–Interval Rough Saw Model | The interval rough and fuzzy evaluations | Environmental image, recycling, pollution control, the environmental management system, environmentally friendly products, resource consumption and green competencies | supplier selection |
| [63] | Rough set theory based hierarchical linear model | Resource-based and Enterprise ecosystem theory | T-test, P-value, error | Grain farms |
| [64] | Framework based on RST | Environmental and Store factors | Frequency, Ranking, Growth rate | Restaurant chain |
| [65] | Generalized attribute reduction in rough set theory | Mean decision power increased attribute reduction (MDPIAR), Positive region preserved attribute reduction (PRPAR) etc. | Micro and Macro evaluation | 16 UCI data sets |
| [66] | Survey of rough set clustering | Variable Precision Model, Total Roughness, Rough K-means | Purity, Entropy | Outliers detection |
| [67] | Rough generation algorithm (RGI) | Rule and tree based classification algorithms | Mean absolute error | Medical data sets |
| [68] | Effective Rough Clustering | ——- | Precision, Accuracy | Super market data set |
| [69] | Rough Set Based Feature Selection | Fuzzy Rough Set Based Feature Selection | A review | Crisp and real-valued data sets |
| [70] | Rough set based decision theory | Decision making by weight | F score, CEI | Reuters Corpus Volume 1 data set |
| [71] | Rough CART algorithm | CART algorithm | Accuracy | Nutrition and health |
| [72] | Rough-Set Feature Selection Model | Decision tree | Error, Accuracy | Survey data |
| [73] | Rough evolutionary algorithm | Evolutionary Algorithm | Courage, Accuracy | Beer preferences, City image data |
| [74] | Foundations of Rough Clustering | Rough k-Means | Lower and upper bounds | Traffic, Web and Supermarket data |
| [75] | Rough set theory | Decision Tree | Rules, Accuracy | Multimedia Data |
| [76] | Rough Self Organizing Map | Crisp clustering | Error, Accuracy | Artificial, Iris data set |
| [62] | Rough Set Theory Fundamental Concepts | Rough Set Theory Principals | Rough Set Theory Data Extraction | Rough Set Theory Applications |
| [77] | Rough classification rules framework | Rough Set theory | Misclassification rate, Accuracy | Interval-valued information system |
| [78] | Rough autonomous Knowledge-Oriented (K-O) clustering | Complete, Single and Average Linkage | Accuracy, Number of clusters | Food nutrient data |
| [29] | Rough set theory | ———— | Rudiments of rough sets | Research directions and applications |

information systems with maximum attribute dependency is chosen by the MDA technique. The MDA technique outperforms its predecessor approaches but itself lacks the generalizability and efficiency. A Variable Precision Rough Set (VPRS) approach utilizes the mean accuracy of approximation to cluster categorical data [5]. The VPRS consider a noisy data and without a predefined clustering attribute it successfully clusters some UCI data sets. Furthermore, the final clusters using divide and conquer method were found comparatively better and are also visualized.

The performance of MMeR in terms of data heterogeneity and uncertainty algorithm was further enhanced by suggesting the Standard Deviation Roughness (SDR) clustering algorithm [80]. The experimental results on certain data sets in terms of cluster purity shows the worth of

SDR as compare to other techniques. Later on, a Standard deviation of Standard Deviation Roughness (SSDR) was introduced in this sequence [81]. The SSDR has the capability to cluster uncertain numerical and categorical data at the same time and hence is proven better than its predecessors like SDR, MMeR and MMR.

Maximum Significance of Attributes (MSA) also computes an appropriate clustering attribute based on the significance of attributes RST concept [32]. The MSA handles the uncertainty and stability for categorical clustering process. The accuracy and purity was also improved up to some extent as compare to MDA, MMR, TR and BC techniques. A clustering technique known as Information-Theoretic Dependency Roughness (ITDR) for categorical data is developed that utilizes the information-theoretic dependencies [33]. A new measure of uncertainty in categorical data was introduced named as information-theoretic entropy. The complexity and purity for the appropriate clustering attribute selection by ITDR was better against SSDR, SDR, MMeR and MMR.

The likelihood function and indiscernibility relation of multivariate multinomial distributions was utilized to develop a novel modified Fuzzy k-Partition method [82]. The idea was effective as it performs extensive theoretical analysis and still achieve lower computational complexity as compare to Fuzzy k-Partition and Fuzzy Centroid approaches. The clustering accuracy and response time were also improved on some real and UCI data. The rough intuitionistic fuzzy K-Mode algorithm was an extension of rough fuzzy k-mode for clustering the categorical data. The parameter of intuitionistic degree in a given cluster was added which calculate the element membership value. The efficiency of suggested scheme as tested on some categorical data of UCI repository which highlights the better results against rough fuzzy k-mode algorithm.

An algorithm called Min-Mean-Mean-Roughness (MMeMeR) was introduced based on enhancements in MMeR and MMR algorithms [83]. A coherent and logical effect of considering the minimum or mean on better accuracy is also analyzed using standard UCI data. They found the objects at edge of a heterogeneous data can be clustered with certainty and are more captivating. Hence, MMeMeR technique was termed effective over existing SDR, MMeR and MMR techniques. Recently, Maximum Value Attribute (MVA) technique is suggested that efficiently cluster the uncertain categorical data [84]. A supplier's data and several UCI data sets are considered to validate the performance of MVA technique with existing approaches. Despite of better performance, it sometimes produce singleton clusters and subject to only domain knowledge. The existing work on rough categorical data clustering is summarized in Table 3.

## 3 An epirical analysis of existing categorical clustering techniques based on Rough Set Theory

Some existing Rough Set based techniques for selecting a clustering attribute in categorical data are analyzed. A well-known technique, Maximum Dependency Attribute (MDA) [31] takes into account the Rough dependency of attributes. The MDA technique chooses best clustering attribute in information system based on maximum dependency degree [87]. Best clustering attribute is selected by MDA technique on the basis of higher dependency degree.

Hassanein and Elmelegy [32] proposes an alternative Rough Clustering Technique known as Maximum Significance Attribute (MSA). In an information system, MSA technique utilizes the significance of attributes. Higher degree of significance in MSA technique determines the best clustering attribute. Though, MDA and MSA techniques perform well in clustering categorical data as compared to their predecessor, they have hardly or sometimes not been able to work on following cases in a categorical data set,

**Table 3. Summary of existing work on rough categorical data clustering.**

| Paper | Proposed Technique | Compared Techniques | Evaluation Metrics | Data Sets/ Application Area |
|---|---|---|---|---|
| [84] | Rough Set based Maximum Value Attribute Technique | K Mean, RST based techniques | Accuracy, Purity, entropy, Time, Iterations | Supplier and UCI data sets |
| [83] | MMeMeR | MMR, MMeR, SDR, Fuzzy K modes, Fuzzy centriods | Accuracy, Purity | Zoo, Soyabean and Mushroom data sets |
| [85] | >Rough intuitionistic fuzzy k-mode | Rough fuzzy k-mode | DB index, D index, XB index, PC pair and Minkowski score | UCI data sets |
| [82] | Modified Fuzzy k-Partition | Fuzzy Centroid and Fuzzy k-Partition | Response time, clustering accuracy | UCI and real data sets |
| [33] | Information Theoretic Dependency Roughness | K-means, Fuzzy K-means, MMR, MMeR, SDR, SSDR | Purity | Zoo data set |
| [86] | Review of categorical Clustering techniques | Min-Min Roughness, Standard Deviation Roughness, Modified Min-Min Roughness, Fuzzy set theory | Uncertainty | Categorical data sets |
| [32] | Maximum Significant Attribute (MSA) | Bi-Clustering, Total roughness, Min-Min Roughness, Maximum Dependent Attribute | Rough accuracy, Purity | Credit card promotion dataset |
| [16] | Variable Precision Model | Total roughness, Min-Min Roughness | Purity, Accuracy | Balloon, Tic-Tac-Toe, SPECT, Hayes-Roth |
| [81] | Standard deviation of Standard Deviation Roughness | Min-Min Roughness, Standard Deviation Roughness, Modified Min-Min Roughness, Fuzzy set theory | Purity | Zoo data set |
| [80] | Standard Deviation Roughness | k-modes, fuzzy k-modes, Min-Min Roughness | Purity | Soybean, Zoo, Mushroom data sets |
| [79] | Modified Min-Min Roughness | Min-Min Roughness, K-Modes, Fuzzy set theory | Purity | Soybean, Zoo, Mushroom data sets |
| [87] | Maximum Dependent Attribute | Bi-Clustering, Total roughness, Min-Min Roughness | Rough accuracy, Iterations | Credit card, Student's qualifications and animal data sets |
| [25] | Min-Min Roughness | Squeezer, K-modes, LCBCDC, ROCK, hierarchical algorithm | Purity | Soybean and Zoo |
| [37] | Total Roughness | Bi- Clustering | Rough accuracy | Small Data sets |

- Independent attributes

- Non-significant attributes

- Equally dependent attributes

- Equally significant attributes

To illustrate these issues, we consider the following example.

**Example 1** Table 4 *is modified data set showing patients with possible viral symptoms* [62]. *There are three conditional attributes: Headache (H), Vomiting (V) and Temperature (T) of six patients. Viral Illness is the decision attribute in* Table 4.

The indiscernibility relation of each attribute induces equivalence classes and considering MDA technique, we calculate the dependency degree of attributes. The dependency degrees of viral data set are given in Table 5. Here, selecting best clustering attribute is not possible as dependency degrees are all equal and 0. Accordingly, the MDA technique fails and hence creates a problem.

In case of MSA technique, we compute the significance of subsets of $U$. The significance degree of all attributes are presented in Table 6. In such situation, the best clustering attribute selection by MSA technique is not possible as all significance values are equal and 0. Therefore, MSA technique also fails and creates a problem.

**Table 4. A Viral Illness information system.**

| Patient | H | V | T | Viral illness |
|---------|---|---|-----------|---------------|
| 1 | 0 | 1 | High | 1 |
| 2 | 1 | 0 | High | 1 |
| 3 | 1 | 1 | Very High | 1 |
| 4 | 0 | 1 | Normal | 0 |
| 5 | 1 | 0 | Normal | 0 |
| 6 | 0 | 0 | Very High | 1 |

**Table 5. Dependency degree of attributes from Table 4.**

| Attribute(depends on) | Degree of Dependency | | MDA |
|-----------------------|---------|---------|-----|
| H | V<br>0 | T<br>0 | 0 |
| V | H<br>0 | T<br>0 | 0 |
| T | H<br>0 | V<br>0 | 0 |

The above example illustrates the lack of ability of existing techniques to deal with zero or equal dependent and significant attributes. Another recent categorical clustering technique ITDR works on the entropy roughness to find clustering attribute [5, 33]. However, entropy is one of the type of purity measure [42] which considers the entire distribution and not just the largest class as it is done by the purity measure [88] in a particular cluster. In other words, the homogeneity or heterogeneity of the cluster does not affect the entropy results [89]. The strength and limitations of existing Rough Set based categorical clustering techniques are highlighted in Table 7. The summary of literature review leading to the proposed research framework is presented as in Fig 1. This figure shows how various researchers contributed towards the main issue of clustering categorical data.

The analysis of existing techniques presented in Table 7 and Fig 1 motivates towards the development of a more comprehensive measure of uncertainty. Accordingly, a measure based on classical information theoretic purity is derived.

## 4 Information-theoretic purity measure with Rough Set Theory

The first and most commonly used purity measures is information gain which is based on Shannon's entropy from information theory [40, 90]. Several variations in classical purity are introduced depending on type of application and a particular uncertainty measurement [39,

**Table 6. Significance degree of attributes from Table 4.**

| Attributes | Significance | | MSA |
|------------|---------|---------|-----|
| H | V<br>0 | T<br>0 | 0 |
| V | H<br>0 | T<br>0 | 0 |
| T | H<br>0 | V<br>0 | 0 |

**Table 7. Strengths and limitations of existing Rough categorical clustering techniques.**

| Technique | Basic idea | Strengths | Limitations |
|---|---|---|---|
| BC | Binary valued attributes | Categorical Data, Uncertainty | Accuracy, Generalization |
| TR | Maximum total roughness | Categorical Data, Complexity, Uncertainty | Purity, Generalization |
| MMR | Maximum mean roughness using lower bound and upper bound | Categorical Data, Complexity, Uncertainty | Purity, Stability, Generalization |
| MDA | Dependency of attributes | Categorical Data, Complexity, Uncertainty | Accuracy, Stability, Purity |
| MSA | Significance of attributes | Categorical Data, Purity, Uncertainty | Stability, Complexity, Entropy |
| ITDR | Information theoretic attribute dependencies | Categorical Data, Purity, Uncertainty | Generalization, Complexity, Accuracy |

41, 42, 89, 91, 92]. In this work, the purity is defined that it can be applied to Rough databases. Hence, the purity of a Rough Set X is illustrated as below.

**Definition 1** In an approximation space $S = (U, Y, V, \xi)$, let $L, M \subseteq Y$ and $L, M \neq \phi$. Rough Purity (RP) of attribute $M$ on attributes $L$, written as $L \Rightarrow_P M$ can be define using by the following equation,

$$
P(M_i|L_j) \quad = \quad
\begin{cases}
|L_j \cap M_i|/|L_j|, & |L_j \cap M_i| > 0 \\
\\
0, & |L_j \cap M_i| = 0
\end{cases}
\tag{1}
$$

Where $P(M_i|L_j)$ is a fuction from $Y$.

**Definition 2** Suppose $y_i \in Y$, $V(y_i)$, has $k$-different values say $\beta_k$, $k = 1, 2, \ldots, n$. Consider a subset of the attributes $M(y_i = \beta_k)$ having $k$-different values of attribute $y_i$. Max-roughness of the set $M(y_i = \beta_k)$ with respect to $y_j$ where $i \neq j$ denoted by $MR_P(M_{i[\gamma]}|L_j)$ as,

$$
MR_P(M_{i[\gamma]}|L_j) = max(P(M|y_i = \gamma)|L_j)
\tag{2}
$$

**Definition 3** $MMR_P(M_i|L_j)$ denotes the Max-mean-roughness of $y_i \in Y$ w.r.t $y_j \in Y$ and is calculated as,

$$
MMR_P(M_i|L_j) = MR_P(M_{i[\gamma]}|L_j) + \ldots + MR_P(M_{i[y_{|V(y_i)|}]}|L_j)|V(y_i)
\tag{3}
$$

$V(y_i)$ is the set of values of attribute $y_i \in Y$ and $i \neq j$.

**Definition 4** Consider number of attributes $a$, max-mean-max-roughness of $y_i \in M$ with respect to $y_j \in L$, where $i \neq j$, refers to the maximum of $MMR_P(y_i|y_j)$, denoted $MMMR_P(M_i|L_j)$ is obtained by the following formula:

$$
MMMR_P(M_i|L_j) = max(MMR_P(M_1|L_1), \ldots, MMR_P(M_m|L_a))
\tag{4}
$$

The Rough Purity Approach (RPA) takes into account the mean degree of Rough Purity to find partitioning attribute. The justification is that the high Rough Purity value implies the more accurate partition attribute is selected. The maximum total roughness of each attribute decides the best crispness [37]. Normally, high purity shows better clustering combination and the clusters are pure subsets of input classes if purity value is high [93].

**Definition 5** To illustrate the computational complexity for RPA technique, let there are $n$ objects, $m$ attributes and $l$ values of each attribute in an information system. The RPA needs $nm$ computation for finding elementary set of all attributes. The Rough Purity of all subsets of $U$ having different values and maximum Rough Purity of all attributes with respect to each other consumes $n^2 l$ computation steps. Accordingly, the steps for finding all mean max-rough purity values are $n$ times. Therefore, the polynomial $O(n^2 l + mn + n)$ comes the computational complexity of RPA.

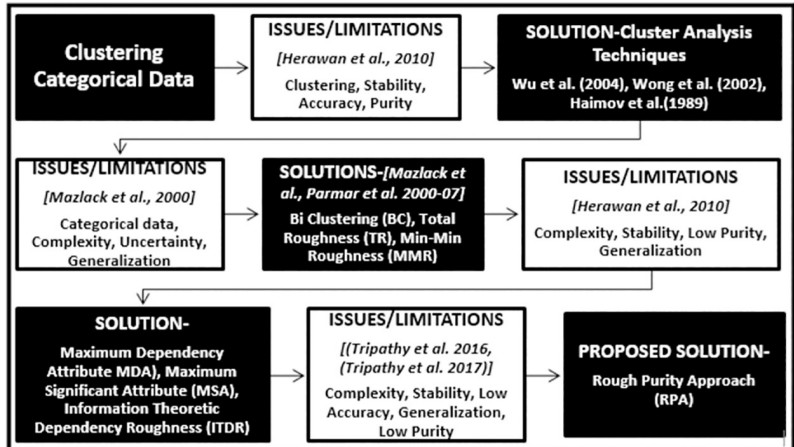

**Fig 1. Scenario leading to the proposed framework.**

The steps involved in RPA technique are presented in Fig 2. Next, we present an illustrative example of the RPA technique.

**Example 2** A student's enrollment qualification information system is presented in Table 8. Degree (D), English (E), Statistics (S), Programming (P) and Mathematics (M) are five categorical attributes of eight students. The best clustering attribute needs to be selected provided no pre-defined decision attribute. For calculating the Rough Purity values, firstly the indiscernibility relations of each attribute must be obtained that induces equivalence classes. Table 8 gives following partitions of object,

1. X(D=B.Sc.)={1, 2}, X(D=M.Sc.)={3, 4, 8},
   X(D=Ph.D.)={5, 6, 7},
   U/D={{1, 2}, {3, 4, 8}, {5, 6, 7}}

## Algorithm RPA

Input: Dataset without clustering attribute
Output: Clustering attribute

1. Compute the equivalence classes using the indiscernibility relation on each attribute.
2. Determine the purity roughness of attribute $a_i$ with respect to all $a_j$, where $i \neq j$.
3. Select information theoretic purity degree of each attribute within attributes.
4. Select the clustering attribute based on maximum degree of rough purity.

**Fig 2. The RPA algorithm.**

**Table 8. Student's enrollment qualification information system.**

| U/A | D | E | S | P | M |
|-----|-----|-----|-----|-----|-----|
| 1 | B.Sc. | Low | No | Fluent | Poor |
| 2 | B.Sc. | Intermediate | Yes | Poor | Fluent |
| 3 | M.Sc. | Advanced | No | Poor | Poor |
| 4 | M.Sc. | Intermediate | No | Fluent | Poor |
| 5 | Ph.D. | Low | Yes | Poor | Fluent |
| 6 | Ph.D. | Advanced | No | Poor | Fluent |
| 7 | Ph.D. | Advanced | Yes | Fluent | Poor |
| 8 | M.Sc. | Advanced | Yes | Fluent | Fluent |

2. X(E=low)={1, 5}, X(E=intermediate)={2, 4},
   X(E=advanced)={3, 6, 7, 8},
   U/E={{1, 5}, {2, 4}, {3, 6, 7, 8}}

3. X(S=no)={1, 3, 4, 6}, X(S=yes) ={2, 5, 7, 8},
   U/S={{1, 3, 4, 6}, {2, 5, 7, 8}}

4. X(P=fluent)={1, 4, 7, 8}, X(P=poor)={2, 3, 5, 6},
   U/P={{1, 4, 7, 8}, {2, 3, 5, 6}}

5. X(M=poor)={1, 3, 4, 7}, X(M=fluent)={2, 5, 6, 8},
   U/M={{1, 3, 4, 7}, {2, 5, 6, 8}}

Definition 4 is used to find the Rough Purity of Degree w.r.t Statistics,
P(S=yes|B.Sc.)=({2, 5, 7, 8}, {1, 2})=1/2=0.5
P(S=yes| M.Sc.)=({2, 5, 7, 8}, {3, 4, 8})=1/3=0.33
P(S=yes | Ph.D)=({2, 5, 7, 8}, {5, 6, 7})=2/3=0.67
P(S=no | B.Sc)=({1, 3, 4, 6}, {1, 2})=1/2=0.5
P(S=no | M.Sc.)=({1, 3, 4, 6}, {3, 4, 8})=2/3=0.67
P(S=no | Ph.D.)=({1, 3, 4, 6}, {5, 6, 7})=1/3=0.33
The maximum roughness degree of Statistics (S) w.r.t Degree (D) can be calculated as:
MP(Syes)=max(0.5,0.33,0.67)=0.67,
MP(Sno)=max(0.5,0.67,0.33)=0.67.
The mean Rough Purity of attribute Statistics (S) with respect to Degree (D) are
MMP(S)=(D$|(L|S = no)$+ D$|(L|S = yes)$) /$|V(D)|$ =(0.67+0.67)/2 = 0.67

Proceeding similarly, each attribute mean Rough Purity is computed. Table 9 summarizes the calculations with RPA, which shows that the high mean purity value is of Mathematics attribute. Considering the heuristic that the high purity shows better clustering combinations, therefore, best clustering attribute is selected as Mathematics. Hence, the clusters obtained are (1,3,4,7), (2,5,6,8).

The comparison of Rough Purity and other measures of uncertainty are illustrated in Propsotion 1.

**Proposition 1** *Rough Purity is more comprehensive measure of uncertainty as compared to Rough Dependency and significance of attributes.*

**Proof:** If the attributes are not dependent on each other, then dependency degree [31] results zero. Similarly, it can be proved that independent attributes are also non-significant. Hence significance of attribute [32] also gives zero. Irrespective of above cases that attributes

**Table 9. MMP roughnes of Table 8.**

|  | Mean Rough Purity | | | | | Mean |
|---|---|---|---|---|---|---|
|  | D | E | S | P | M |  |
| D | - | 0.5 | 0.4167 | 0.4167 | 0.4167 | 0.4375 |
| E | 0.5567 | - | 0.333 | 0.333 | 0.333 | 0.3889 |
| S | 0.667 | 0.5 | - | 0.5 | 0.75 | 0.604 |
| P | 0.667 | 0.5 | 0.5 | - | 0.75 | 0.604 |
| M | 0.667 | 0.5 | 0.75 | 0.75 | - | **0.667** |

are not dependent or they are not significant for each other, the Rough Purity measure will always give a non zero value. In other words, the Eq 2 always gives,

$$MR_P(M_{i[\gamma]}|L_j) = max(P(M|y_i = \gamma)|L_j) \neq 0 \qquad (5)$$

Hence, it is proved that Rough Purity is more comprehensive measure of uncertainty than Rough Dependency and significance of attributes.

## 5 Experimental setup and data sets description

RPA technique is validated using C#. The results are presented in form of tables. The domain of Supplier Base Management (SBM) is used to validate the proposed RPA technique [43]. SBM data set comprises ten attributes (shown in Table 10) showing performance information and supplier capability of 23 Suppliers (S). The attribute included are Quality Management Practices and systems ($Q_m$), Documentation and Self-audit ($D_s$), Process/manufacturing Capability ($P_c$), Management of Firm ($M_f$), Design and Development Capabilities ($D_c$), Cost (C), Quality (Q), Price (P), Delivery (D), Cost Reduction Performance ($C_p$) and Others (O). The efficiency of each supplier is determined by applying the Data Envelopment Analysis [43]. The last column of Table 10 shows their conclusion on each supplier. The domain of all attributes contain continuous value because the categorical data is already normalized.

RPA technique is also validated using six data sets taken from UCIML repository. They includes: Balloons (16 instances, 4 attributes), Car Evaluation (1728 instances, 6 attributes), Zoo (101 instances, 17 attributes) and Chess (3196 instances, 37 attributes), Balance scale (625 instances, 5 attributes), Monk's problems (432 instances, 8 attributes). RPA is tested with all these data sets and compared with recent Rough Categorical techniques MDA, MSA and ITDR on basis of various evaluation measures like Time, Iterations, Purity, Entropy, Accuracy and Rough Accuracy.

## 6 Results and discussion

Table 11 illustrates the time complexity of MDA, MSA, ITDR and RPA techniques to complete the clustering task. For Balloons data set, the number of instances are less therefore the response time is same for all techniques. Moreover, RPA takes lesser time as compared to all techniques for Car, Zoo and Chess data sets.

The iterative complexity depends on number of attributes and attribute values of a data set. It also includes the steps like finding dependency degree of all attributes for MDA, maximum significance of all possible combinations of attributes for MSA, minimum Rough Entropy for ITDR and maximum Rough Purity for RPA. It can also be seen from Table 12 that the RPA consumes minimum iterations all data set than the MDA and MSA techniques. According to Table 12, despite the fact that the RPA and ITDR techniques undergo almost similar iterative

**Table 10. Discretized supply base management data set.**

| S | $Q_m$ | $D_s$ | $P_c$ | $M_f$ | $D_c$ | C | Q | P | D | $C_p$ | O | E |
|---|---|---|---|---|---|---|---|---|---|---|---|---|
| 1 | 3 | 2 | 3 | 3 | 4 | 2 | 1 | 1 | 1 | 1 | 1 | I |
| 2 | 2 | 2 | 1 | 1 | 1 | 2 | 2 | 1 | 1 | 1 | 1 | E |
| 3 | 1 | 1 | 1 | 1 | 2 | 1 | 3 | 1 | 4 | 1 | 3 | E |
| 4 | 5 | 2 | 2 | 2 | 3 | 4 | 5 | 2 | 4 | 1 | 4 | E |
| 5 | 5 | 2 | 3 | 3 | 4 | 4 | 3 | 3 | 3 | 1 | 2 | I |
| 6 | 3 | 2 | 2 | 3 | 3 | 3 | 3 | 3 | 4 | 2 | 3 | E |
| 7 | 2 | 1 | 3 | 2 | 4 | 3 | 4 | 2 | 2 | 2 | 3 | E |
| 8 | 5 | 2 | 3 | 3 | 3 | 3 | 2 | 2 | 2 | 1 | 2 | I |
| 9 | 5 | 2 | 3 | 3 | 4 | 4 | 1 | 1 | 1 | 1 | 1 | I |
| 10 | 1 | 2 | 1 | 1 | 1 | 1 | 5 | 1 | 4 | 1 | 3 | E |
| 11 | 2 | 1 | 1 | 2 | 3 | 2 | 1 | 1 | 3 | 1 | 2 | I |
| 12 | 1 | 1 | 3 | 3 | 3 | 3 | 3 | 3 | 2 | 1 | 2 | E |
| 13 | 5 | 2 | 3 | 3 | 4 | 4 | 1 | 2 | 3 | 1 | 3 | I |
| 14 | 4 | 2 | 3 | 3 | 4 | 4 | 2 | 1 | 2 | 1 | 2 | I |
| 15 | 4 | 2 | 2 | 3 | 1 | 1 | 4 | 3 | 4 | 2 | 3 | E |
| 16 | 3 | 2 | 3 | 3 | 2 | 2 | 3 | 1 | 1 | 1 | 1 | I |
| 17 | 5 | 2 | 3 | 3 | 4 | 3 | 3 | 1 | 2 | 1 | 1 | I |
| 18 | 5 | 2 | 3 | 3 | 4 | 4 | 4 | 1 | 1 | 1 | 1 | I |
| 19 | 4 | 2 | 3 | 1 | 4 | 3 | 4 | 1 | 4 | 1 | 3 | I |
| 20 | 4 | 2 | 3 | 3 | 1 | 4 | 2 | 2 | 4 | 1 | 4 | E |
| 21 | 5 | 2 | 3 | 2 | 4 | 4 | 2 | 1 | 3 | 1 | 2 | I |
| 22 | 5 | 2 | 2 | 3 | 3 | 4 | 5 | 3 | 4 | 2 | 4 | E |
| 23 | 4 | 2 | 3 | 3 | 2 | 3 | 4 | 3 | 3 | 2 | 4 | E |

complexity to get their best clustering attribute but the RPA has still the better time taken. The reason is Rough Purity formula is computationally simpler than Rough Entropy therefore the effect can be seen on response time. The relevant induced indiscernibility relation will show the clusters obtained by selected best attribute.

Table 13 shows the performance of RPA, MDA, MSA and ITDR techniques in terms of Purity, Entropy, Accurracy and Rough Accuracy. The achieved accuracy on all data sets as presented in Table 13 shows that the proposed RPA technique outperformed other techniques except Balance Scale and Monk's Problem where the accuracy is the same. Similarly, Table 13 also illustrates the entropy of obtained clusters by each technique. Less entropy shows better

**Table 11. Time complexity of all techniques.**

| Data Set | Response Time (millisec) | | | |
|---|---|---|---|---|
| | MDA | MSA | ITDR | RPA |
| **Balloons** | 0 | 0 | 0 | **0** |
| **Car Evaluation** | 20 | 595 | 17 | **15** |
| **Zoo** | 6 | 116 | 8 | **6** |
| **Chess** | 31598 | 658068 | 815 | **800** |
| **Balance scale** | 2 | 27 | 5 | **2** |
| **M's Problem** | 4 | 72 | 3 | **3** |
| **SBM** | 1 | 15 | 3 | **1** |

**Table 12. Iterative complexity of all techniques.**

| Data Set | Minimum iterations | | | |
|---|---|---|---|---|
| | **MDA** | **MSA** | **ITDR** | **RPA** |
| **Balloons** | 80 | 147 | 49 | **25** |
| **Car Evaluation** | 3519 | 12138 | 367 | **184** |
| **Zoo** | 4381 | 23461 | 1201 | **600** |
| **Chess** | 781127 | 3892358 | 5181 | **2591** |
| **Balance scale** | 624 | 1404 | 301 | **150** |
| **M's Problem** | 1397 | 4218 | 239 | **120** |
| **SBM** | 660 | 2779 | 1373 | **680** |

clustering technique [45] and it can be seen from this table that the proposed technique shows lesser entropy for all data sets except Balance Scale and Monk's Problem where entropy is the same. Hence, RPA performance is better for entropy measure too. Moreover, the purity of obtained clusters by each technique as presented in Table 13 shows that the RPA technique has better purity for all data sets except Car evaluation, Balance Scale and Monk's Problem data set where all techniques produce equal purity of their best clustering attribute. Finally, Table 13 presents the Rough Accuracy of the techniques. The reason of less or zero Rough Accuracy value is that this measure is not a comprehensive measure of uncertainty [34]. The overall performance of RPA technique in terms of Rough accuracy is still better as compared to other techniques.

If two or more techniques select similar clustering attributes then the evaluation measures produced by those techniques are also similar. For example in case of Monk's problems data set, the MDA and MSA techniques select similar clustering attribute hence their Accuracy, Purity and Entropy values are similar. Similarly, for the same data set the ITDR and RPA choose the same attribute as best. Despite the fact that these techniques can choose same best

**Table 13. Comparative performance of techniques for all data sets.**

| Measure | Technique | Balloons | Car Evaluation | Zoo | Chess | Balance Scale | Monks Problem | SBM |
|---|---|---|---|---|---|---|---|---|
| **Purity** | **RPA** | **0.8** | **0.7** | **0.61** | **0.6** | **0.64** | **0.5** | **0.74** |
| | MDA | 0.6 | 0.7 | 0.59 | 0.52 | 0.64 | 0.5 | 0.61 |
| | MSA | 0.6 | 0.7 | 0.4 | 0.54 | 0.64 | 0.5 | 0.74 |
| | ITDR | 0.6 | 0.7 | 0.5 | 0.52 | 0.64 | 0.5 | 0.74 |
| **Entropy** | **RPA** | **0.16** | **0.29** | **0.43** | **0.28** | **0.36** | **0.3** | **0.22** |
| | MDA | 0.29 | 0.33 | 0.5 | 0.3 | 0.36 | 0.3 | 0.29 |
| | MSA | 0.29 | 0.33 | 0.7 | 0.3 | 0.36 | 0.3 | 0.23 |
| | ITDR | 0.29 | 0.3 | 0.48 | 0.3 | 0.36 | 0.3 | 0.22 |
| **Accuracy** | **RPA** | **0.66** | **0.53** | **0.72** | **0.52** | **0.6** | **0.5** | **0.6** |
| | MDA | 0.47 | 0.48 | 0.55 | 0.5 | 0.6 | 0.5 | 0.5 |
| | MSA | 0.47 | 0.48 | 0.5 | 0.5 | 0.6 | 0.5 | 0.55 |
| | ITDR | 0.47 | 0.5 | 0.69 | 0.5 | 0.6 | 0.5 | 0.6 |
| **Rough Accuracy** | **RPA** | **0.2** | **0.1** | **0.2** | 0 | 0 | 0 | **0.11** |
| | MDA | 0 | 0 | 0.1 | 0 | 0 | 0 | 0 |
| | MSA | 0 | 0 | 0.1 | 0 | 0 | 0 | 0.1 |
| | ITDR | 0.1 | 0 | 0.1 | 0 | 0 | 0 | 0.11 |

**Table 14. Average percentage improvement of time by RPA technique.**

|  | MDA | MSA | ITDR | RPA |
|---|---|---|---|---|
| **Average Time (milisec)** | 4518.714 | 94127.57 | 121.5714 | 118.1429 |
| **Improvement by RPA** | 97.40% | 99.87% | 2.82% |  |

**Table 15. Average percentage improvement of iterations by RPA technique.**

|  | MDA | MSA | ITDR | RPA |
|---|---|---|---|---|
| **Average Iterations** | 113112.6 | 562357.9 | 1244.429 | 621.4286 |
| **Improvement by RPA** | 99.45% | 99.88% | 50.06% |  |

**Table 16. Average percentage improvement of Purity by RPA technique.**

| Technique | Average Purity | Improvement by RPA |
|---|---|---|
| MDA | 0.59 | 11.14% |
| MSA | 0.59 | 11.14% |
| ITDR | 0.60 | 9.30% |
| **RPA** | 0.655714 |  |

clustering attribute, but the number of iterations, time taken and hence, the complexity is still promising for RPA technique as the data sets size increases.

## 7 Summary of results

This section summarizes the average percentage improvement and overall percentage improvement by RPA technique for clustering categorical data as compared to MDA, MSA, and ITDR. This summary of results shows that the RPA technique significantly improves the time, iterations, purity, entropy, and accuracy. Table 14 shows a slight response time improvement by RPA against ITDR but as compared to MSA and MDA techniques, the percentage improvement is large. It is also observed in Table 15, that the RPA techniques require almost half iterations as compared to ITDR and 100% fewer iterations against MDA and MSA techniques to choose the best clustering attribute. Similarly, the Tables 16–18 clearly show the significant improvement in terms of several clustering evaluation measures like purity, entropy and accuracy by RPA technique against MDA and MSA technique. Though the ITDR technique outperforms MDA and MSA for these measures but still the performance of RPA is reasonably improved as compared to ITDR technique for clustering the categorical data. Finally, Table 19 highlights the comparative overall improvement by RPA in terms of Time, Iterations, Purity, Entropy, and Accuracy. It can be clearly seen that RPA technique not only proved to be less complex but also more efficient in selecting the best clustering attribute and clustering categorical data. Hence, it can be summarized from the whole experimental results that the

**Table 17. Average percentage improvement of entropy by RPA technique.**

| Technique | Average Entropy | Improvement by RPA |
|---|---|---|
| MDA | 0.338571 | 13.92% |
| MSA | 0.358571 | 18.72% |
| ITDR | 0.321429 | 9.33% |
| **RPA** | 0.291429 |  |

**Table 18. Average percentage improvement of accuracy by RPA technique.**

| Technique | Average Accuracy | Improvement by RPA |
|-----------|------------------|---------------------|
| MDA | 0.5143 | 14.72% |
| MSA | 0.5143 | 14.72% |
| ITDR | 0.5514 | 7% |
| **RPA** | 0.59 | |

**Table 19. Overall percentage improvement by RPA technique.**

| Measure | Overall Improvement by RPA |
|---------|----------------------------|
| Time | 66.70% |
| Iterations | 83.13% |
| Purity | 10.53% |
| Entropy | 14% |
| Accuracy | 12.15% |

proposed RPA technique is not only simple, more generalized, and quick but also more perfect clusters were obtained having less entropy and high purity and accuracy.

## 8 Threats to validity

The primary threat to validity for this study is that the tools of existing approaches like MDA, MSA and ITDR are not available, they are re-implemented via a prototype implementation system. This system is developed using C# for experimental purpose. However, our code of previous approaches is strictly based on the descriptions and pseudo codes available in their respective research articles. To reduce the influence of this biasness and as remedy, similar data sets and same evaluation measures were considered as used by other existing techniques. As a result, it is ensured that all evaluation measures of existing techniques give the same results as computed in their original work.

Another threat to validity for this study is related to the number of instances and attributes of dataset. In this study, a real SBM and six bench mark data sets were chosen for experiments. Moreover, to generalize our results, it was necessary to perform experiments with data sets of various number of instances and attributes. Accordingly, the data sets considered for experimentation were chosen from different application domains. However, this study only focused on small and medium size data sets. Experiments on large data sets may be performed to further validate the proposed technique.

## 9 Conclusion

The traditional clustering techniques are not able to deal with uncertainty in the data set as they are not designed to do so. Several categorical data clustering techniques have emerged as a new trend in techniques of handling uncertainty in the clustering process. The motivation of a better Rough Clustering technique is developed after exposing some potential issues of recently developed Rough Clustering techniques like MDA, MSA and ITDR. These issues include data with attributes having zero or equal dependency, attributes with zero or equal significance value and random attribute selection. The key contribution of this paper is that these limitations of existing rough set based clustering techniques for categorical data are handled successfully and effectively. A Rough set based information theoretic approach for clustering

categorical data with uncertainty named Rough Purity (RPA) approach is hence presented. The extensive experimental analysis of the proposed RPA and existing approaches using a supplier base management real data set and UCI benchmark data sets are discussed. The significant improvement can be seen in experimental outcomes in terms of relevant parameters like time (66.70%), iterations (83.13%), purity (10.53%), entropy (14%), accuracy (12.15%) and rough accuracy of clusters. This significant improvement by the proposed technique shows that RPA can be extended for further research in the field of Data Mining, Artificial Intelligence, Rough Set Theory and soft computing etc. One of the limitation of this research is that the analyses of only relevant rough set based categorical techniques like MDA, MSA and ITDR is presented. Though, this comparison provides strong evidence about the efficiency of the proposed approach in terms of several evaluation parameters, but other approaches like fuzzy bipolar soft set and Pythagorean fuzzy bipolar soft set etc. need to be compared to further analyze the RPA technique.

## Author Contributions

**Conceptualization:** Jamal Uddin, Rozaida Ghazali, Jemal H. Abawajy, Habib Shah, Noor Aida Husaini, Asim Zeb.

**Data curation:** Jamal Uddin.

**Formal analysis:** Jamal Uddin, Jemal H. Abawajy, Noor Aida Husaini, Asim Zeb.

**Funding acquisition:** Rozaida Ghazali, Habib Shah.

**Investigation:** Jamal Uddin.

**Methodology:** Jamal Uddin, Rozaida Ghazali, Jemal H. Abawajy, Noor Aida Husaini.

**Project administration:** Jamal Uddin, Jemal H. Abawajy, Habib Shah, Noor Aida Husaini, Asim Zeb.

**Resources:** Jamal Uddin, Rozaida Ghazali, Jemal H. Abawajy, Noor Aida Husaini.

**Software:** Jamal Uddin, Noor Aida Husaini.

**Supervision:** Rozaida Ghazali, Jemal H. Abawajy, Noor Aida Husaini, Asim Zeb.

**Validation:** Jamal Uddin.

**Visualization:** Jamal Uddin.

**Writing – original draft:** Jamal Uddin.

**Writing – review & editing:** Jamal Uddin, Rozaida Ghazali, Jemal H. Abawajy, Habib Shah, Noor Aida Husaini, Asim Zeb.

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
