## [Decision Letter · Decision Letter 0]

2 Dec 2021

PONE-D-21-21593Rough Set Based Information Theoretic Approach for Clustering Uncertain Categorical DataPLOS ONE

Dear Dr. GHAZALI,

Thank you for submitting your manuscript to PLOS ONE. After careful consideration, we feel that it has merit but does not fully meet PLOS ONE’s publication criteria as it currently stands. Therefore, we invite you to submit a revised version of the manuscript that addresses the points raised during the review process.

ACADEMIC EDITOR: The reviewer has asked for revisions. There is concerns about the discussion and the comparisons, that authors need to address. Based on all this, I am recommending major revisions. 

Furthermore when submitting the revised paper, please also consider the following points:

1. English language needs proofreading.

2. References should be in proper format.

3. All acronyms must first be defined.

We look forward to receiving your revised manuscript.

Kind regards,

Usman Qamar

Academic Editor

PLOS ONE

Journal Requirements:

[The authors would like to thank the King Khalid University of Saudi Arabia for 

supporting this research under grant number R.G.P.1/365/42.]

 [The authors would like to thank the King Khalid University of Saudi Arabia for supporting this research under grant number R.G.P.1/365/42.]

Reviewers' comments:

Reviewer's Responses to Questions

**Comments to the Author**

1. Is the manuscript technically sound, and do the data support the conclusions?

Reviewer #1: Partly

2. Has the statistical analysis been performed appropriately and rigorously? 

Reviewer #1: Yes

3. Have the authors made all data underlying the findings in their manuscript fully available?

Reviewer #1: Yes

4. Is the manuscript presented in an intelligible fashion and written in standard English?

Reviewer #1: No

5. Review Comments to the Author

Reviewer #1: The paper is well-written in general and the authors have done a good job of communicating their ideas.

Your abstract does not highlight the specifics of your research or findings. There needs to be an explicit research objective stated, preferably as a separate section.

The related work section should be extended to present a critical review of existing techniques highlighting their deficiencies. At present, the literature review is presented in a "this did that" format. There is no flow here. This section should be re-written based on the techniques rather than listing the papers. It is better to provide a tabular format summary of the existing approaches along with strengths and weaknesses as well.

For the methodology, it was explained clearly, however it should be supported with an example. The algorithm proposed in the paper has no formal proof that will produce the correct score.

Analyses are missing using more state-of-the-art methods. Compare and provide strong evidence about the efficiency of the proposed approach with similar approaches based on fuzzy bipolar soft set and Pythagorean fuzzy bipolar soft set, so far, no such experimentation is provided. Show the robustness checking of the proposed model. How the proposed approach is effective in terms of computational resources like memory and execution time.

The conclusion is not precise. The key findings and its implementation potential (in practice) is missing. Clearly, identify its academic contributions also. Limitations are not mentioned.

6. PLOS authors have the option to publish the peer review history of their article (what does this mean?). If published, this will include your full peer review and any attached files.

Reviewer #1: No

---

## [Author Response · Author response to Decision Letter 0]

25 Jan 2022

ACADEMIC EDITOR: 

When submitting the revised paper, please also consider the following points:

Comment: 1. English language needs proofreading.

Response: English proof reading is done and as per capability all relevant issues are resolved.

Comment: 2. References should be in proper format.

Response: References are reviewed to remove any mistake.

Comment: 3. All acronyms must first be defined.

Response: Acronyms are defined at first place.

JOURNAL REQUIREMENTS:

Comment: 1. Please ensure that your manuscript meets PLOS ONE's style requirements, including those for file naming. The PLOS ONE style templates can be found at 

Response: All above requirements are considered in preparing the revised manuscript.

[The authors would like to thank the King Khalid University of Saudi Arabia for 

supporting this research under grant number R.G.P.1/365/42.]

Comment: 2. We note that you have provided funding information that is currently declared in your Funding Statement. However, funding information should not appear in the Acknowledgments section or other areas of your manuscript. We will only publish funding information present in the Funding Statement section of the online submission form. 

Please remove any funding-related text from the manuscript and let us know how you would like to update your Funding Statement. 

Currently, your Funding Statement reads as follows: 

 [The authors would like to thank the King Khalid University of Saudi Arabia for supporting this research under grant number R.G.P.1/365/42.]

Response: The funding-related text is now removed from the manuscript.

Comment: 3. Please include a separate caption for each figure in your manuscript.

Response: The caption of Figures is now included as per instructions.

REVIEWERS' COMMENTS:

Reviewer #1: The paper is well-written in general and the authors have done a good job of communicating their ideas.

Comment: Your abstract does not highlight the specifics of your research or findings. There needs to be an explicit research objective stated, preferably as a separate section.

Response: The Abstract is rearranged and rephrased accordingly. The Motivation, Problem Statement, Objectives, Methods, Results and Conclusion is now explicitly stated as separate sections.

 Comment: The related work section should be extended to present a critical review of existing techniques highlighting their deficiencies. At present, the literature review is presented in a "this did that" format. There is no flow here. This section should be re-written based on the techniques rather than listing the papers. It is better to provide a tabular format summary of the existing approaches along with strengths and weaknesses as well.

Response: The Related work section (Section 2, Subsections 2.1, 2.2 & 2.3) is now presented as per the valuable suggestion of respected reviewer. The tabular summary is also provided accordingly (Table 1, 2 & 3). The strength and weakness of relevant existing approaches are highlighted in Section 3, Example 1 and Table 7.

For the methodology, it was explained clearly, however it should be supported with an example. The algorithm proposed in the paper has no formal proof that will produce the correct score.

Response: Example 2 illustrate the methodology of proposed approach. It presents how successfully the clusters can be obtained using the proposed approach.

Analyses are missing using more state-of-the-art methods. Compare and provide strong evidence about the efficiency of the proposed approach with similar approaches based on fuzzy bipolar soft set and Pythagorean fuzzy bipolar soft set, so far, no such experimentation is provided.

Response: With all due respect, the goal of this research is to explore some significant limitations in existing rough set based categorical clustering techniques only. Therefore, particularly the cases where these techniques are unable to produce quality clusters (independent and insignificant data) are considered. Accordingly, we suggest a better, viable and more comprehensive alternative approach RPA in our research. Therefore, the analyses of only relevant rough set based categorical techniques like MDA, MSA and ITDR is presented in Section 6, Table 11-19. This comparison provides strong evidence about the efficiency of the proposed approach in terms of several evaluation parameters. 

The comparison with other approaches like fuzzy bipolar soft set and Pythagorean fuzzy bipolar soft set is valuable point but due to scope limitations we have mention it as future work in Section 9. 

Show the robustness checking of the proposed model. How the proposed approach is effective in terms of computational resources like memory and execution time.

Response: The computational complexity of proposed clustering strategy is determined by the number of iterations required and in terms of respond time. It is mentioned in 2nd last paragraph of Section 1 (Introduction). The computational complexity in terms of Big O is illustrated in Definition 5. Table 11, 12, 14 & 15 shows the relevant experimental results.

The conclusion is not precise. The key findings and its implementation potential (in practice) is missing. Clearly, identify its academic contributions also. Limitations are not mentioned.

Response: The Conclusion (Section 9) is now rearranged to include key findings, contributions and limitations. Other limitations are highlighted in Section 8.

While revising your submission, please upload your figure files to the Preflight Analysis and Conversion Engine (PACE) digital diagnostic tool, https://pacev2.apexcovantage.com/. 

Response: The Figures are now enhanced using PACE. Thanks for anther valuable suggestion.

---

## [Decision Letter · Decision Letter 1]

28 Feb 2022

Rough Set Based Information Theoretic Approach for Clustering Uncertain Categorical Data

PONE-D-21-21593R1

Dear Dr. GHAZALI,

We’re pleased to inform you that your manuscript has been judged scientifically suitable for publication and will be formally accepted for publication once it meets all outstanding technical requirements.

Kind regards,

Usman Qamar

Academic Editor

PLOS ONE

Additional Editor Comments (optional):

Reviewers' comments:

Reviewer's Responses to Questions

**Comments to the Author**

1. If the authors have adequately addressed your comments raised in a previous round of review and you feel that this manuscript is now acceptable for publication, you may indicate that here to bypass the “Comments to the Author” section, enter your conflict of interest statement in the “Confidential to Editor” section, and submit your "Accept" recommendation.

Reviewer #1: (No Response)

2. Is the manuscript technically sound, and do the data support the conclusions?

Reviewer #1: (No Response)

3. Has the statistical analysis been performed appropriately and rigorously? 

Reviewer #1: (No Response)

4. Have the authors made all data underlying the findings in their manuscript fully available?

Reviewer #1: (No Response)

5. Is the manuscript presented in an intelligible fashion and written in standard English?

Reviewer #1: (No Response)

6. Review Comments to the Author

Reviewer #1: Authors addressed the previous comments and the paper has been substantially improved. I vote for its acceptance.

7. PLOS authors have the option to publish the peer review history of their article (what does this mean?). If published, this will include your full peer review and any attached files.

Reviewer #1: No

---

## [Editor Report · Acceptance letter]

11 Apr 2022

PONE-D-21-21593R1 

Rough Set Based Information Theoretic Approach for Clustering Uncertain Categorical Data 

Dear Dr. Ghazali:

I'm pleased to inform you that your manuscript has been deemed suitable for publication in PLOS ONE. Congratulations! Your manuscript is now with our production department. 

Kind regards, 

on behalf of

Dr. Usman Qamar 

Academic Editor

PLOS ONE